# Heat Shock Proteins in Pancreatic Cancer: Pathogenic Mechanisms and Clinical Implications

**DOI:** 10.3390/cells14201627

**Published:** 2025-10-18

**Authors:** Jacek Kabut, Jakub Sokołowski, Wiktoria Żelazna, Mateusz Stępień, Marta Strauchman, Natalia Jaworska, Jakub Wnuk, Anita Gorzelak-Magiera, Łukasz Michalecki, Iwona Gisterek-Grocholska

**Affiliations:** 1Department of Oncology and Radiotherapy, Medical University of Silesia, 40-615 Katowice, Polandanitagor@op.pl (A.G.-M.); igisterek@sum.edu.pl (I.G.-G.); 2Student Scientific Club, Department and Clinic of Oncology and Radiotherapy, The Faculty of Medical Sciences, Medical University of Silesia, 40-615 Katowice, Polandwikzel575@gmail.com (W.Ż.); nataliajaworska@op.pl (N.J.)

**Keywords:** heat shock proteins, pancreatic cancer, clinical value

## Abstract

Heat shock proteins (HSPs) are highly conserved molecular chaperones that play a key role in maintaining protein homeostasis, or proteostasis, especially under stressful environmental conditions such as hyperthermia, hypoxia, or the presence of reactive oxygen species. In pancreatic cancer, the expression of many HSP isoforms is dysregulated, contributing to the activation of mechanisms that promote tumor development, including proliferation, invasion, angiogenesis, treatment resistance, and cancer cachexia syndrome. HSPs are significant diagnostic and prognostic biomarkers. Some of them, such as HSP27, HSP70, and HSP90, have been shown to correlate with treatment response and patient survival. Others, including HSPA2 and HSPB6, may indicate an increased risk of disease recurrence. These proteins also represent promising therapeutic targets. Preclinical and clinical studies suggest that inhibiting HSP activity and associated signaling pathways may inhibit tumor growth and increase treatment efficacy. These therapeutic effects include inducing apoptosis, autophagy, and ferroptosis, as well as sensitizing cancer cells to chemotherapy and immunotherapy. This article summarizes the current knowledge about the role of HSPs in pancreatic cancer biology, their significance as biomarkers, and their potential therapeutic applications in treating pancreatic ductal adenocarcinoma (PDAC). Most studies conducted so far have been preclinical, and due to the promising results, further clinical investigation is warranted.

## 1. Introduction

Pancreatic ductal adenocarcinoma (PDAC) is considered one of the most aggressive cancers with increasing rates of incidence and mortality. It is estimated that PDAC was responsible for 466,003 deaths in 2020,which puts it as the seventh leading cause of cancer-related deaths and will be the second-most common cause of death among oncological patients in the USA by 2030 [1,2]. In the European Union, the number of deaths caused by PC is predicted to increase to 111,500 by 2025, which will make PC the third leading cause of cancer-related deaths in the EU [3].

The risk factors of PC include obesity, diabetes mellitus, a diet rich in processed meat, lack of physical activity, and tobacco smoking. Gene mutations such as *p53*, *K-ras*, *p16*, and *PRSS1* contribute to an increased risk of pancreatic cancer [4,5].

Pancreatic ductal adenocarcinoma is the most common histopathological subtype of PC, which accounts for more than 90% of PC diagnoses. The other types of PC are considered to have an impact on prognosis but have a lesser impact on treatment decisions [6].

In the early stage of the disease, pancreatoduodenectomy is the choice of treatment in a resectable disease. The role of neoadjuvant treatment, which includes systemic treatment with multi-drug regimens or chemoradiation, remains uncertain in the case of the resectable disease, borderline resectable, and locally advanced disease. Systemic adjuvant treatment in PDAC with mFOLFIRINOX regimen is recommended for all eligible patients. In metastatic PDAC, multi-drug regimens such as FOLFIRINOX, gemcitabine with nab-paclitaxel, or monotherapy with gemcitabine are recommended in systemic therapy [7].

The FOLFIRINOX regimen was compared to gemcitabine in monotherapy in the Conroy study, which included advanced PDAC without previous history of treatment. The median time of overall survival in the group of patients treated with the FOLFIRINOX regimen was 11.1 months, compared to 6.8 months in the gemcitabine group [8].

In a study comparing gemcitabine in monotherapy to gemcitabine accompanied by nab-paclitaxel, OS was 6.7 months compared to 8.5 months in the two-drug regimen [9].

Limited effectiveness of the systemic approach in PDAC treatment might be caused by the microenvironment surrounding the growing tumor. Desmoplastic response of surrounding tissues and low angiogenesis are the causes of inadequate chemotherapy effects [10].

Despite the progress in diagnosis and treatment, PDAC remains a disease with poor survival rates. Even with radical treatment including surgical approach and adjuvant systemic therapy, the 5-year relative survival rate is 11.5%, and it is largely due to late diagnosis and asymptomatic course at early stages [11,12]. Therefore, the search for novel therapies is still a valid task.

HSPs are a large family of conserved proteins found in both prokaryotic and eukaryotic cells [13]. Their key functions include helping to fold newly synthesized polypeptides, refolding denatured or unstable proteins, facilitating intracellular transport, and preventing the formation of toxic protein aggregates [14]. This is why they are referred to as “molecular chaperones” [15]. The functional diversity of heat shock proteins arises from their capacity to perform multiple roles, including those of foldases, holdases, sequestrases, aggregases, and disaggregases [16,17,18,19,20]. Under normal conditions, the level of HSP synthesis is similar to that of other proteins. However, in response to cellular stress, such as elevated temperature, hypoxia, infection, oxidative stress, DNA damage, or the accumulation of improperly folded proteins, there is a significant increase in HSP expression [13,21]. This mechanism depends on the activation of transcription factors from the HSF family. After oligomerization, these factors bind to heat shock elements (HSEs) located within the promoter regions of HSP genes [15]. Dysregulation of HSP expression occurs in many cancers. These proteins have been shown to regulate the proliferation, apoptosis, invasion, and metastasis of cancer cells. They also play a role in developing resistance to chemotherapy and radiotherapy [22].

HSPs are classified by molecular weight into six main families: HSP100, HSP90, HSP70, HSP60, HSP40, and HSP20, which are low-molecular-weight HSPs [13,15]. These groups each perform specific functions that affect cellular homeostasis and tumor transformation processes differently. Figure 1 presents a summary of HSP roles.

## 2. Aims

This study aims to critically analyze the role of heat shock proteins in diagnosing, predicting, and treating pancreatic cancer, with a focus on their potential use as biomarkers and therapeutic targets. The current state of knowledge regarding the significance of individual HSP families in the pathogenesis of pancreatic ductal cancer was discussed. Experimental data and clinical observations were considered to highlight the molecular mechanisms of HSP action and their role in disease progression, treatment response, and therapeutic resistance. We also analyzed therapeutic strategies based on HSP inhibition and their potential application in cancer treatment. This review focuses exclusively on heat shock proteins in pancreatic ductal adenocarcinoma (PDAC). Premalignant lesions (PanIN, IPMN) and inflammatory conditions (e.g., pancreatitis) were not included in our systematic search or synthesis.

## 3. Materials and Methods

A literature search on pathogenic mechanisms and clinical implications regarding heat shock proteins in pancreatic cancer was conducted using the PubMed, Scopus, and Google Scholar databases. Relevant keywords and associated phrases were applied: “Heat Shock Proteins”, “HSP”, “Pancreatic Ductal Adrenocarcinoma”, “PDAC Immunotherapy”, “HSPs Role In PDAC”, and “HSP prognostic factor”.

The types of studies analyzed included original articles, systematic reviews, and meta-analyses. The language of the studies was primarily English. Studies were published within the period from 2005 to 2025.

In order to prepare a summary of clinical trials involving heat shock protein inhibitors, the following databases were searched: ClinicalTrials.gov, WHO International Clinical Trials Registry Platform (ICTRP), and the EU Clinical Trials Register. The search combined the terms “pancreatic cancer,” “pancreatic ductal adenocarcinoma,” and “PDAC” with the names of known HSP inhibitors: Tanespimycin (17-AAG),Alvespimycin (17-DMAG), IPI-504 (retaspimycin), Ganetespib (STA-9090), Luminespib (AUY922), Onalespib (AT13387),XL888, PU-H71, SNX-5422, MKT-077, Pifitrin-μ (PES, 2), JG-98, JG-231, VER-155008,Apatorsen (OGX-427), RP101 (brivudine/brom vinyl-deoxyuridine, BVDU), PEN-866, andTAS-116 (Pimitespib).

The details of the included studies and clinical trials are presented in Appendix A.

## 4. The Biological and Prognostic Significance of HSP Families in Pancreatic Ductal Adenocarcinoma (PDAC)

### 4.1. Low-Molecular-Weight Heat Shock Protein (lmHSPs)

Low-molecular-weightheat shock proteins (lmHSPs) play a key role in preventing the aggregation of improperly folded proteins, which constitutes their most important protective function [15]. They are found mostly in muscles including the heart muscle. They contain a fragment of amino acids in the C-terminal region, known as the α-crystallin domain [23]. Furthermore, they exhibit antiapoptotic action and support the processes of angiogenesis [24,25]. The Hsp20 family includes ten proteins (HspB1–HspB10) [21], of which particular significance is attributed to Hsp27 (HspB1), which performs regulatory functions in the scope of apoptosis and the cell cycle [22,24,26].

Low-molecular-weight heat shock proteins (lmHSPs), including HSP27, HSPB2, HSPB6, and αA-crystallin, play a complex role in the biology of pancreatic cancer by exhibiting both pro- and anti-tumorigenic effects. Studies conducted on pancreatic cancer cell lines and cancer-associated fibroblasts (CAFs) have shown that increased HSPB6 expression in CAFs may contribute to the modulation of the tumor stroma. Moreover, analysis of data from the Clinical Proteomic Tumor Analysis Consortium (CPTAC) revealed that high HSPB6 expression correlates with longer overall survival in patients with PDAC, indicating its potential as a prognostic marker [27].

In pancreatic cancer cell line models, HSP27 has been shown to regulate the expression of Snail, E-cadherin, and ERCC1, thereby potentially contributing to the development of resistance to gemcitabine. Lower levels of HSP27 are associated with poorer prognosis, while elevated expression correlates with improved responses to chemotherapy, suggesting its role as a predictive biomarker of treatment efficacy [22,28].

Deng et al. (2010) [29] demonstrated that the small heat shock protein αA-crystallin is physiologically expressed in normal pancreatic tissue and acts as a negative regulator of pancreatic tumorigenesis. Their study revealed a significant downregulation of αA-crystallin in pancreatic tumor tissue compared to adjacent non-neoplastic tissue. Functional assays further confirmed that overexpression of αA-crystallin in pancreatic cancer cells reduced tumorigenic potential, whereas its silencing enhanced tumorigenicity [29].

### 4.2. HSP40 (DnaJ Family)

The Hsp40 family, comprising 49 proteins, is the largest group of heat shock proteins, divided into three subgroups: DnaJA, DnaJB, and DnaJC. These proteins act as co-chaperones for Hsp70, inducing their ATPase activity thanks to their J-domain [21,22,30]. Hsp40s decide the fate of Hsp70 substrates and help in protein folding, assembly, and degradation.

The J-domain serves to stabilize Hsp70 complexes with improperly folded proteins while stimulating the activity of Hsp70 [31]. A representative of Hsp40 family is Hsp47, which plays a key role in the biosynthesis of collagen [32].

HSPs from the HSP40 family, also referred to as DnaJ proteins, play a crucial role in regulating the function of HSP70 chaperones and are increasingly recognized as contributors to PDAC progression by modulating cancer cell survival, metabolic reprogramming, and apoptosis.

Liu et al. demonstrated that DnaJB11, a co-chaperone of HSPA5 (BiP, Grp78), is overexpressed in pancreatic cancer cells. Their study, which was primarily based on in vitro assays and in vivo mouse xenograft models implanted with human PDAC cells, showed that increased DnaJB11 levels promote tumor growth and inhibit apoptosis, suggesting a tumor-supportive role for this co-chaperone in the PDAC microenvironment [33].

Similarly, Roth et al. provided further evidence for the involvement of HSP40 family members in PDAC pathogenesis. Their research, conducted on established PDAC cell lines, revealed that elevated expression of DnaJA1 enhances the Warburg effect, upregulates antiapoptotic Bcl-2 protein levels, and reduces apoptotic signaling, thereby promoting tumor cell survival and invasiveness [34]. These findings, although limited to preclinical models, highlight the potential of HSP40 proteins as modulators of cancer metabolism and apoptosis in pancreatic cancer.

### 4.3. HSP60 (Chaperonins)

Hsp60s are proteins that function mainly in mitochondria, helping to maintain their integrity and being responsible for proper ATP-dependent folding of substrates [15,35,36,37,38]. The isolated form of Hsp60, Tailess Complex Peptide (TCP/TRiC), functions in the cytosol [38]. The activity of these proteins is also maintained under conditions of no cellular stress [39].

HSP60, a mitochondrial chaperonin, plays a multifaceted role in PDAC by regulating protein homeostasis, cellular metabolism, and apoptotic signaling. Elevated HSP60 expression has been observed in PDAC tissues and is associated with enhanced tumor cell proliferation and resistance to cell death. Mechanistically, HSP60 supports mitochondrial integrity and oxidative phosphorylation, contributing to cancer cell survival under metabolic stress. Additionally, HSP60 may inhibit apoptotic pathways by stabilizing antiapoptotic proteins and preventing cytochrome c release, thereby promoting PDAC progression and chemoresistance [37].

### 4.4. HSP70

The Hsp70 family is divided into two main subgroups: DnaK-like (canonical Hsp70) and Hsp110, which includes four isoforms: HspH1, HspH2, HspH3, and Grp170 (glucose-regulated protein 170) [36,40]. These proteins take part in the folding of newly formed polypeptides and the repair of damaged protein conformation during cellular stress [15,21,35]. Hsp110, by cooperating with Hsp70, also plays a significant role in disassembling amyloid aggregates. It helps to counteract neurodegenerative diseases [41,42]. Hsp70 proteins also perform a protective function against stress-induced apoptosis and support angiogenesis [43].

HSP70 plays a critical role in the progression of pancreatic cancer, affecting both tumor biology and the systemic condition of patients. In a study conducted on pancreatic cancer cell lines, Liumei et al. demonstrated that elevated HSP70 expression activates the NF-κB signaling pathway and promotes epithelial–mesenchymal transition (EMT), thereby enhancing the proliferation, migration, and invasiveness of cancer cells [43]. In contrast, Zhai et al., using tumor samples derived from pancreatic cancer patients, confirmed that increased expression of the HSPA2 gene—also present in stromal components—correlates with a more aggressive clinical course [44]. HSP70, which is released in extracellular vesicles by PDAC cells, has been shown to activate the p38β MAPK catabolic cascade, contributing to muscle wasting and the development of cancer cachexia. This mechanism was demonstrated in study [45] using patient-derived pancreatic cancer cells, while its systemic effects were further confirmed in vivo in a murine model of cancer cachexia [46]. Notably, HSP70 has been identified as an independent prognostic factor for both overall and progression-free survival. Studies by Xiong et al. indicate that analyzing HSP70 and VEGF levels simultaneously may improve the accuracy of predicting responses to chemotherapy and radiotherapy; a decrease in these levels after treatment was associated with a more favorable prognosis [43]. These findings were obtained in clinical studies conducted in patients.

### 4.5. HSP90

The Hsp90 family proteins are ATP-dependent molecules. They possess ATPase activity, whereby ATP binding and hydrolysis have a significant impact on the conformational dynamics of Hsp90 [47]. Hsp90s play a key role in protein stabilization. They prevent aggregation and take part in cellular signal transduction. They are involved in the functioning of many proteins, including transcription factors, steroid receptors, epidermal growth factor receptors (EGFR, HER2), and TLR4 (Toll-like receptor 4) [15,21,48,49]. Additionally, Hsp90s influence the activity and degradation of many oncoproteins, which is of particular importance in oncological processes [50]. The Hsp90 family includes several isoforms: GRP94, which is located in the endoplasmic reticulum, TRAP1 in the mitochondria, and HSP90α and HSP90β in the cytosol [51,52,53,54]. Studies using murine pancreatic cancer cell lines have shown a significant upregulation of HSP90 expression in PDAC cells [55].

Clinical studies have identified HSP90 as a potential prognostic biomarker in humans. Elevated HSP90 levels were associated with a threefold increased risk of mortality, particularly in patients with a history of acute pancreatitis, independently of other clinical variables [56].

HSP90 also holds promise in imaging diagnostics. Its expression can be visualized using positron emission tomography (PET) tracers such as ^64^Cu-Di-San A1 and ^18^F-PEGylated San A. However, these tracers may yield false-positive signals in areas of inflammation. To improve specificity, a novel PET probe, ^18^F-NOTA-Dimer-San A, was developed and validated in murine models, allowing for precise detection of HSP90 expression in malignant tissues while distinguishing it from inflammatory lesions [53,54,57].

A summary of the role of HSPs in pathogenesis and biology in PDAC is presented in Table 1.

## 5. The Role of HSP in Treating Pancreatic Cancer

HSPs, including HSP27, HSP47, HSP60, HSP70, and HSP90, are essential molecular chaperones involved in the maintenance of proteostasis under physiological and pathological conditions. In PDAC, their overexpression is closely associated with aggressive tumor behavior, therapeutic resistance, and immune evasion. Elevated levels of these chaperones correlate with advanced disease stages, poor prognosis, and decreased responsiveness to standard therapies, thus highlighting their potential as actionable molecular targets. The therapeutic efficacy of HSP inhibition appears to be dependent on the tumor’s specific genetic background. For example, mortalin (HSPA9), a mitochondrial HSP70 family member, is significantly upregulated in KRAS-mutated PDAC. Silencing mortalin in such contexts induces apoptosis and increases mitochondrial membrane permeability. Preclinical investigations have demonstrated that JG-231, a hydrophilic derivative of the HSP70 inhibitor MKT-077, effectively suppresses tumor growth in PDAC models harboring the KRASG12C mutation, suggesting a promising therapeutic approach tailored to molecular tumor profiles [65].

### 5.1. HSP27: Marker of Resistance and Therapeutic Target

HSP27 contributes to PDAC progression and metastasis via activation of the β-catenin/MMP-3 axis. Retrospective clinical studies indicate that elevated HSP27 expression correlates with advanced tumor stage and poor prognosis [66]. These associations, however, have not been clinically validated in interventional trials. Preclinical models—primarily in vitro and mouse xenografts—demonstrate that HSP27 knockdown using siRNA or small-molecule inhibitors such as OGX-427 enhances FOLFIRINOX efficacy and mitigates phosphorylation-related resistance mechanisms [67]. Additionally, gemcitabine-induced accumulation of methylglyoxal (MG) has been shown to trigger HSP27 expression as a cytoprotective response [68,69]. Compounds like triptolide, AHCC, and melatonin have been reported to downregulate HSP27 and restore apoptosis in PDAC cells, with melatonin exerting its effects via inhibition of NF-κB and STAT3 signaling, as demonstrated in experimental in vitro studies [70,71].

In the study by Drexler et al., lower HSP27 expression was associated with shorter overall survival (OS). Furthermore, high expression was associated with a better response to the gemcitabine regimen in patients with resectable, non-metastatic disease [72].

### 5.2. HSP47: Modulator of Tumor Microenvironment

HSP47 facilitates extracellular matrix (ECM) remodeling, creating a physical barrier that impedes drug penetration. Its inhibition improves gemcitabine sensitivity, indicating its relevance in targeting the tumor microenvironment [73,74]. These findings are based on experimental in vitro and in vivo studies using PDAC cell lines and mouse xenograft models. Han et al. presented a strategy to increase drug delivery to the tumor site. In their study, they used a tumor microenvironment-responsive nanosystem based on PEGylated polyethylenimine-coated gold nanoparticles to deliver all-trans retinoic acid (ATRA) and siRNA targeting heat shock protein 47. This influences the activated pancreatic stellate cells (PSCs) and inhibits extracellular matrix hyperplasia [75].To date, no clinical trials in humans have evaluated therapeutic strategies directly targeting HSP47 in pancreatic cancer.

### 5.3. HSP60: Regulator of Mitochondrial Metabolism and Tumor Immunogenicity

HSP60 has been implicated in the progression of PDAC by promoting cancer cell proliferation, migration, and tumorigenic potential. Zhou et al. demonstrated that HSP60 expression is significantly elevated in PDAC tissues and correlates with tumor progression. Analysis of patient-derived pancreatic cancer cells revealed that HSP60 exerts its oncogenic effects through stabilization of mitochondrial oxidative phosphorylation (OXPHOS) and modulation of the HSP60/OXPHOS/Erk1/2 signaling axis. This pathway supports tumor cell survival by maintaining mitochondrial function and sustaining Erk1/2 phosphorylation. Inhibition of OXPHOS—either by genetic silencing of HSP60 or pharmacologically via metformin—leads to reduced Erk1/2 activation, induction of apoptosis, and cell cycle arrest [37]. Additionally, HSP60 interacts with antiapoptotic proteins such as Bcl-xL, survivin, and clusterin, further enhancing its role in resistance to cell death. Functional studies confirmed that HSP60 knockdown suppresses PDAC cell proliferation and invasiveness, whereas its overexpression accelerates tumor progression. Additionally, thermal stress induced by local hyperthermia in the range of 39–43 °C leads to increased surface expression of HSP60 and HSP70 on cancer cells, which enhances their antigenic profile. This promotes antigen presentation by dendritic cells and augments antitumor immune responses through interferon-gamma (IFN-γ) secretion by activated T cells [76]. These findings are based primarily on experimental in vitro and in vivo models. Although local hyperthermia is used clinically as an adjunct to cancer therapy, there is currently no direct clinical evidence confirming that this approach enhances HSP-mediated immunogenicity or IFN-γ-driven immune responses in patients.

### 5.4. HSP70: Multifaceted Therapeutic Target

HSP70 is markedly overexpressed in pancreatic PDAC and is closely associated with increased tumor proliferation, apoptosis resistance, invasiveness, and poor prognosis [77]. Due to its multifunctional role in cancer progression, HSP70 has emerged as a promising therapeutic target. Pharmacological inhibition of HSP70 suppresses tumor growth and enhances the efficacy of chemotherapy and immunotherapy. Small-molecule inhibitors such as PES, MKT-077, VER-155008, and Ap-4-139B have demonstrated preclinical efficacy, with the latter showing synergistic antitumor activity when combined with hydroxychloroquine in murine models of metastatic PDAC [77].

Natural compounds including ursenolide and maslinic acid preferentially target glucose-deprived PDAC cells by inhibiting HSPA5 (GRP78) and GRP94, inducing endoplasmic reticulum stress in vitro [57]. Additionally, maslinic acid suppresses HSPA8 and promotes autophagy, although its overexpression may limit treatment efficacy, underscoring the need for combination strategies targeting multiple HSP70 isoforms [78].

Beyond cytoprotection, HSP70 inhibition elicits immunomodulatory effects by activating dendritic cells and enhancing antitumor immune responses, as demonstrated in in vitro and in vivo models [79]. HSP70 also stabilizes the oncogenic mutant p53 R175H protein, and its inhibition promotes degradation of this variant, attenuating tumor progression [80]. Moreover, HSPA5 facilitates ferroptosis in PDAC cells via EP300-mediated acetylation, with HDAC6 acting as a negative regulator—suggesting potential synergy through dual inhibition [81].

Combination therapies targeting HSP70 show promise. Leja-Szpak et al. demonstrated that gemcitabine combined with melatonin or AFMK enhances apoptotic signaling in PANC-1 cells more effectively than monotherapy by downregulating HSP70 and cIAP-2 [82]. Similarly, radiofrequency ablation (RFA) increases HSP70 expression and activates the AKT–mTOR axis, promoting survival; however, its combination with mTOR inhibitors achieves a synergistic antitumor effect in murine PDAC models [83]. These findings are based on preclinical studies; although RFA is used clinically, the described molecular effects remain unconfirmed in humans.

HSP70 also contributes to PDAC metastasis through interaction with its co-chaperone STIP1, which stabilizes the HSP70–HSP90 complex and activates the FAK/AKT/MMP signaling pathway. High STIP1 expression correlates with poor prognosis, and its inhibition reduces migration and invasion in experimental models [84]. Additionally, miR-634 acts as a tumor suppressor by targeting HSPA2, inhibiting epithelial-to-mesenchymal transition and extracellular matrix degradation. Clinical sample analysis supports an inverse correlation between miR-634 and HSPA2 levels, while in vitro assays confirm that miR-634 restoration suppresses malignant traits [84]. These findings remain limited to preclinical studies and have not yet been validated in clinical trials.

The glucose-regulated proteins (GRPs) are Ca^2+^-binding chaperone proteins with protective properties whose transcription is induced in response to several stimuli that disrupt ER structure and function, related to HSP70. In Park’s study, the novel therapeutic agents PST-A and PST-B exhibited selective cytotoxicity against PANC-1 pancreatic cancer cells under glucose deprivation. This was attributed to the inhibition of glucose-regulated protein 78 (GRP78), a heat shock protein (HSP) that protects pancreatic cancer cells [85]. Another novel strategy was presented in the study by Tang et al. Secalonic acid D was found to inhibit the Akt signaling pathway and affect the induction of glucose-regulated protein 78 (GRP78) under glucose-starved conditions, resulting in a cytotoxic effect on human pancreatic carcinoma PANC-1 cells [86]. Figure 2 summarizes the role of the HSP70 family in PDAC.

### 5.5. HSP90: Central Regulator of Oncogenic Stability

HSP90 is a central molecular chaperone that stabilizes a wide array of oncogenic client proteins, including EGFR, HER2, VEGF, phosphorylated STAT3, Src, and IGF-1Rβ. Through this activity, it supports key hallmarks of PDAC, such as sustained proliferation, invasion, angiogenesis, and resistance to therapy [87,88,89,90]. Pharmacological inhibition of the HSP90 ATPase domain using agents such as ganetespib, 17-AAG (tanespimycin), and AUY922 (luminespib) promotes proteasomal degradation of these client proteins, enhances sensitivity to chemotherapy and radiotherapy, and disrupts DNA repair mechanisms by downregulating ATM, ATR, RAD51, and DNA-PK [90]. Although these compounds have shown potent antitumor activity in preclinical models, early-phase clinical trials in PDAC patients—such as those evaluating 17-AAG in combination with gemcitabine—have yielded mixed results, and further clinical validation is needed to confirm therapeutic efficacy.

A secreted isoform, HSP90α, also contributes to PDAC progression via paracrine mechanisms. It binds to the LRP1 (CD91) receptor, activating the AKT signaling cascade and inducing epithelial-to-mesenchymal transition (EMT). In preclinical studies, neutralization of extracellular HSP90α using the monoclonal antibody HH01 reversed EMT and suppressed metastatic potential [91,92]. Moreover, small-molecule inhibitors that disrupt the HSP90–Cdc37 interaction (e.g., x6506 and x1540) have been shown to inhibit ERK and AKT signaling in KRAS-mutated PDAC cells [93].

HSP90 also cooperates with HSP70 to regulate the stability and membrane localization of SLC6A14, an amino acid transporter frequently overexpressed in PDAC. Inhibition of HSP90 destabilizes SLC6A14 and reduces amino acid uptake, while combination with SLC6A14 antagonists such as α-methyl-tryptophan enhances antitumor effects in vivo [94].

Resistance to therapy in PDAC is often driven by compensatory activation of survival pathways. Thiadiazole-based HSP90 inhibitors have been shown to overcome such resistance by destabilizing oncogenic proteins and inhibiting the PI3K/AKT/mTOR axis. Co-administration with MEK inhibitors results in robust tumor growth inhibition and prolonged survival in murine models [95,96].

Within the immunosuppressive tumor microenvironment, HSP90 plays an additional role by stabilizing STAT1 and promoting IFN-γ-induced upregulation of immune checkpoint molecules such as PD-L1 and immunomodulatory enzymes including IDO1. Pharmacological inhibition of HSP90 using agents such as luminespib, ganetespib, SNX-2112, or XL888 decreases the expression of these immunosuppressive markers and enhances the efficacy of immune checkpoint blockade, including anti–PD-1 therapies, in preclinical PDAC models [97,98].

Lastly, iron oxide nanoparticles (DIO-NPs) have been shown to trigger oxidative stress in PDAC cells, leading to upregulation of HSP70 and HSP90 as part of a cytoprotective response. This stress adaptation can be effectively counteracted by co-treatment with HSP inhibitors, thereby amplifying the overall antitumor effect both in vitro and in vivo [99]. Figure 3 presents a summary of the role of the HSP90 in PDAC.

Table 2 provides an overview of the diagnostic and prognostic implications of HSPs in PDAC, while Table 3 outlines their potential as therapeutic targets.

## 6. Clinical Translation Challenges of HSP Inhibition

Preclinical studies clearly point to HSPs as promising therapeutic targets in PDAC, but the step from laboratory findings to clinical benefit is far from straightforward. HSPs are not tumor-specific proteins; they are essential components of proteostasis in many normal tissues, particularly those with high turnover such as the intestinal epithelium, bone marrow precursors, or neurons. As a result, systemic inhibition of HSP70 or HSP90 has repeatedly been associated with adverse events, including liver, kidney, and hematologic toxicity, which in several early clinical trials became the main reason for discontinuing treatment.

Another challenge is biological rather than toxicological. High HSP expression in pancreatic tumors may reflect a stress-adapted phenotype rather than a true oncogenic driver. Blocking HSP activity in such a context could be bypassed by other stress-response programs, for example, unfolded protein response, autophagy, or antioxidant pathways, which reduce the durability of the therapeutic effect.

Finally, it is becoming increasingly clear that not all PDACs are equally dependent on HSP function. Identifying which patients are most likely to benefit from HSP inhibition will require stratification strategies that take into account the mutational background (e.g., KRAS variants), dominant stress signatures, or isoform-specific HSP expression patterns. Approaches that combine HSP analysis with angiogenic or immune markers may be particularly informative. Altogether, these considerations argue for biomarker-driven, carefully designed trials and for combining HSP inhibitors with other targeted agents in order to improve efficacy while keeping systemic toxicity within acceptable limits.

## 7. Heat Shock Protein Inhibitors in Clinical Trials

Due to promising preclinical results, many heat shock protein inhibitors have been qualified for clinical trials, including pancreatic cancer trials. HSP90 inhibitors are the most frequently studied molecules in this field. However, due to insufficient clinical activity, excessive toxicity, or economic constraints, experimental drugs for PDAC have only reached Phase I and II trials.

### 7.1. HSP27 Inhibitors

Fahring and co-author conducted a series of trials using the molecule RP101 [(E)-5-(2-bromovinyl)-2′-deoxyuridine (BVDU)], which preclinically demonstrated proapoptotic activity and chemioresistance inhibition in pancreatic cancer cell lines. The first study enrolled 13 patients with pancreatic cancer (including 4 stage-III and 9 stage-IV tumors). A combined therapy with Gemcitabine, Cisplatin, and RP101 was conducted. The median overall survival (OS) was 447 days, and the response rate (RR) was 33.In the second trial, 21 similar patients were recruited, while Cisplatin was excluded from treatment. A total of 83% of the patients survived over six months, and 33% over one year. No adverse events (AEs) related to RP101 were reported [101].

Due to the promising results, a Phase II trial was conducted. The study enrolled 168 pancreatic cancer patients (25% stage III, 75% stage IV). However, the median OS was higher in the placebo group than in the gemcitabine and RP101 cohort. The authors also highlight the unexpectedly long survival of the placebo group compared to similar trials (8.87 months). Furthermore, patients with a Body Surface Area (BSA) > 1.85 were found to benefit from RP101 therapy, while patients with a lower BSA reported significantly more serious AEs than the corresponding placebo cohort [102].

An additional HSP27 inhibitor is the antisense nucleotide Apatorsen (OGX-427). In the Phase II trial, 132 patients with metastatic pancreatic cancer were divided into 66-patient cohorts. The median OS was 5.3 and 6.9 for the treatment and placebo groups, respectively. Additionally, the following AEs were reported: nausea (*n* = 8), diarrhea (*n* = 7), fatigue (*n* = 5), vomiting (*n* = 5), anemia (*n* = 2), thrombocytopenia (*n* = 2), and leukopenia (*n* = 1), all of them mostly grade 1/2. The authors also highlighted that the subjects with initially high serum HSP27 levels benefited more from Apatorsen treatment [103].

### 7.2. HSP90 Family Inhibitors

Pacey and colleagues conducted a Phase I study of alvespimycin (17-DMAG)—a geldanamycin derivative. Two of the twenty-five subjects enrolled were pancreatic cancer patients. No clinical activity was observed in this cohort, but the drug was well tolerated at a dose of <80 mg/m^2^. The most common AEs included nausea, vomiting, fatigue, and liver enzyme disturbances, which were low-grade and reversible. Four patients developed 10 visual change AEs, and at a dose >106 mg/m^2^, serious adverse events (SAEs) were reported, including one treatment-related death [104].

Another molecule being studied is ganetespib (STA-9090). Fifty-three subjects were enrolled in the Phase I study, including two pancreatic cancer patients. The participants were divided into three dose-based cohorts and demonstrated a sufficient safety profile to initiate Phase II trial. Concurrently, all patients experienced at least one AE: diarrhea (88.7%), fatigue (56.6%), abdominal pain (37.7%), nausea (34%), and increased ALP (18.9%), AST (17%), and ALT (11.3%). In addition, 15% reported visual changes, and one subject with coronary artery disease developed grade I atrioventricular block. Anemia and neutropenia each occurred in 5.7% of the patients [105].

The Phase II study included 14 patients with refractory metastatic pancreatic cancer. Disease Control Rate (DCR) was achieved in 28.6% of subjects, median OS was 4.57 months, and median progression-free survival (PFS) was 1.58 months. Due to the suboptimal clinical effect of monotherapy, the authors suggested combining ganetespib with standard chemotherapy [106].

Tanespimycin (17-AAG) is another geldanamycin derivative being investigated. In Phase I study (13), 30 participants were enrolled, including 1pancreatic cancer patient. AEs included G3 diarrhea (*n* = 3), which was considered Drug-Limiting Toxicity (DLT); G3 or G4 hepatotoxicity (elevated AST and/or ALT, *n* = 1); vomiting (considered a DLT); and G3 and G2 hypersensitivity reactions (*n* = 3) [107].

Pedersen and co-authors conducted a Phase II study enrolling 21 patients with stage IV pancreatic adenocarcinoma. Due to the lack of complete or partial response, and a median OS of 5.4 months, they concluded that the increased toxicity associated with Gemcitabine and 17-AAG combined therapy outweighed the benefits. Furthermore, 65% of AEs were G3, including nausea (15%), vomiting (15%), dehydration (10%), constipation (10%), anorexia (10%), lymphopenia (10%), and leukopenia (10%). Additionally, 15% of AEs were G4 (neutropenia, lymphopenia) [108].

Luminespib (AUY-922) had two Phase I trials. The first examined the safety of combining the HSP90 inhibitor with Capecitabine in patients with advanced solid tumors. Of the 23 subjects, 4were pancreatic cancer patients. Two of them achieved stable disease status at the end of the study. The most common AEs again included the following: vision changes (78%), diarrhea (61%), and fatigue (43%). The only SAE was hand–foot syndrome. One patient experienced a DLT of G3 diarrhea [109].

A similar study was conducted by Doi et al., who examined the effects of AUY-922 in the Japanese population. Of the 31 patients, 2 had pancreatic cancer, and the recorded AEs were similar to those in the previous study [110].

Finally, Renouf et al. enrolled 12 subjects with metastatic or locally advanced pancreatic adenocarcinoma after at least one line of treatment in a Phase II study. Unfortunately, only one patient developed stable disease, and nine developed a progressive disease. The median OS was only 2.9 months. G3 AEs included fatigue (8%) and AST elevation (8%), while milder and more common toxicity incorporated ECG QT interval prolongation (50%), diarrhea (33%), and anemia (33%) [111].

Among the newer drugs being tested is PEN-866. The Phase I/IIa trial enrolled 21 patients, including patients with pancreatic adenocarcinoma. The most common AEs (>20% of subjects) were nausea, diarrhea, fatigue, vomiting, alopecia, and neutropenia. Ultimately, the drug was found to have an acceptable safety profile [112].

In 2020, the results of the Phase Ib trial of XL-888 in combination with pembrolizumab were published, which enrolled 14 participants with gastrointestinal tumors, including 5pancreatic adenocarcinoma patients. One DLT was reported—G3 autoimmune hepatitis—and other treatment-related toxicities were retinopathy (G2; *n* = 2), nausea (G2; *n* = 1), constipation (G2; *n* = 1), and diarrhea (G2; *n* = 3). An acceptable safety profile was also documented in this case [113].

### 7.3. Current Clinical Trials

Despite the relatively acceptable safety profile of HSP inhibitors, due to their limited clinical benefits, they have not been qualified for Phase III trials in patients with pancreatic cancer. To our knowledge, the only currently recruiting study is the Phase I trial (3) of AB122. Although the molecule itself is not an HSP inhibitor, one of the cohorts (B-1), consisting of patients with PDAC, is scheduled to receive AB122 and pimitespib (TAS-116, HSP90 inhibitor) combined treatment. The analyzed parameters will include DLT, AEs, ORR, and 6-month PFS.

## 8. Summary

This paper provides a comprehensive overview of current knowledge regarding the importance of heat shock proteins (HSPs) in pancreatic cancer pathogenesis, diagnosis, and treatment, with a focus on their potential as biomarkers and therapeutic targets. Numerous scientific reports confirm HSP participation in fundamental carcinogenic processes, including tumor growth, invasion, metastasis, tumor microenvironment remodeling, apoptosis avoidance, cachexia development, and systemic treatment and radiotherapy resistance. Interestingly, some HSP isoforms, such as HSPB2, exhibit anti-cancer properties, e.g., activating the p53 protein and limiting PDAC cell proliferation. From a translational perspective, HSPs show significant potential as diagnostic and prognostic biomarkers in pancreatic cancer. For example, increased HSPB6 expression correlates with a more favorable prognosis in patients with pancreatic ductal adenocarcinoma. In contrast, decreased HSP27 levels are associated with unfavorable clinical parameters, such as poor histopathological differentiation, more frequent liver metastases, and shorter survival after tumor resection. Simultaneously assessing HSP70 and VEGF levels is a promising method for predicting treatment response. HSPA2 and HSP90, on the other hand, have been identified as unfavorable prognostic factors associated with a higher risk of disease recurrence and shorter overall survival. Additionally, HSP90 is used in molecular imaging as a target for radiolabeled ligands in positron emission tomography (PET); however, its clinical use is limited due to its rapid metabolism and elimination by the hepatobiliary system.

Heat shock proteins are important therapeutic targets for treating pancreatic cancer. Preclinical studies have demonstrated that inhibiting these proteins can make cancer cells more susceptible to chemotherapy, modulate the immune response by affecting PD-L1 and IDO1 expression, and induce direct cytotoxic effects. Substances such as melatonin, metformin, AFMK, ganetespib, and JG-231 demonstrate antitumor activity in PDAC models by inducing apoptosis, autophagy, and ferroptosis and by inhibiting epithelial–mesenchymal transformation processes and amino acid metabolism. Recently, there has been growing interest in combination therapy strategies, which combine HSP inhibitors with MEK and mTOR pathway inhibitors or epigenetic drugs. This approach can significantly increase therapy efficacy by affecting multiple tumor resistance mechanisms simultaneously. Although preclinical results are promising, many HSP-targeted therapies require clinical validation. Further, well-designed randomized studies are required. Nevertheless, mounting evidence suggests that heat shock proteins are an essential component of tumor biology and may play a pivotal role in future therapeutic strategies for pancreatic cancer.

### Future Directions

Future research should focus on the isoform-specific and spatial characterization of HSP families in PDAC to define actionable molecular profiles. Integrating HSP expression with the immune and stromal contexture (e.g., PD-L1, IDO1, CAFs, myeloid cells) may guide rational immunotherapy combinations. The validation of circulating HSPs, auto-antibodies, and HSP-targeted PET tracers could establish non-invasive biomarkers for monitoring disease. Early-phase studies with HSP70/HSP90 inhibitors combined with MEK/mTOR, DNA repair, or ferroptosis-targeting agents are also warranted, ideally supported by predictive biomarkers of response. In parallel, stromal-directed approaches (such as HSP47 inhibition) and targeting extracellular HSP70/90 to counteract cachexia may open new therapeutic opportunities. Finally, assay harmonization (IHC methods, scoring, cut-offs) is needed to improve reproducibility and translation into clinical practice.

## Figures and Tables

**Figure 1 cells-14-01627-f001:**
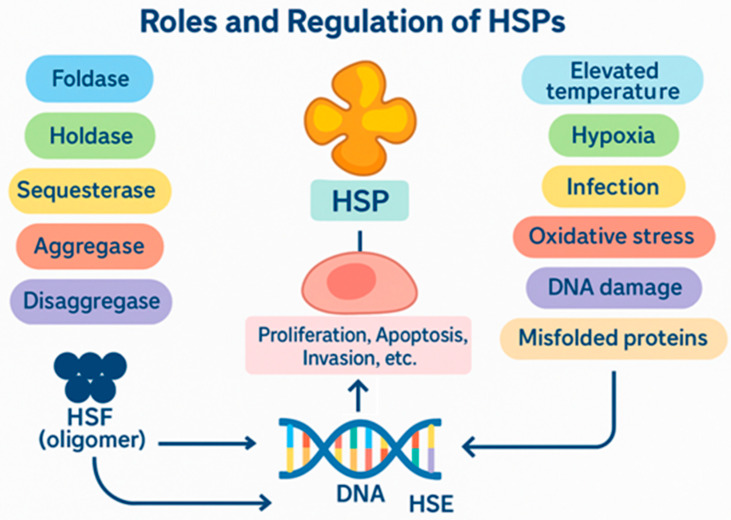
Roles and regulation of heat shock proteins (HSPs)**.** A schematic representation of the activation pathways and functions of HSPs in the cell. The figure illustrates how stressful factors, such as elevated temperature, hypoxia, infection, oxidative stress, DNA damage, and the presence of misfolded proteins, activate HSP synthesis. The transcriptional factor HSF (heat shock factor), in its oligomeric form, binds to the DNA sequence called HSE (heat shock element), which leads to increased expression of HSP genes. Once activated, HSPs perform various roles in the cell, including foldase (protein folding), holdase (maintaining protein conformation), sequestrase (isolation), aggregase (aggregation), and disaggregase (disassembling aggregates). Research on HSP signaling pathways is conducted both in vitro and in vivo.

**Figure 2 cells-14-01627-f002:**
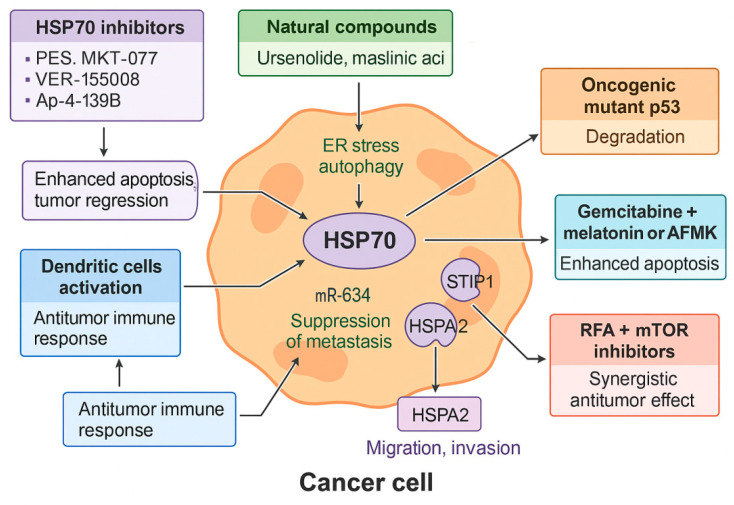
The role of HSP70 in PDAC and therapeutic intervention strategies. This schematic diagram illustrates the multifaceted role of Heat Shock Protein 70 in a cancer cell and various therapeutic strategies targeting it. In the center, HSP70 is shown to support key pathological processes, including ER stress autophagy and the suppression of metastasis by modulating miR-634. Its interaction with HSPA2 and STIP1 (stress-induced phosphoprotein 1) promotes cancer cell migration and invasion. The figure also highlights how HSP70 contributes to the degradation of oncogenic mutant p53. The diagram presents several therapeutic interventions: HSP70 inhibitors (PES, MKT-077, VER-155008, Ap-4-139B) lead to enhanced apoptosis and tumor regression. Natural compounds (urseolide, maslinic acid) also target this pathway. Combined treatments like gemcitabine + melatonin or AFMK (N1-acetyl-N2-formyl-5-methoxykynuramine) and RFA (radiofrequency ablation) + mTOR (mammalian Target of Rapamycin) inhibitors demonstrate synergistic antitumor effects. The activation of dendritic cells, a result of HSP70 inhibition, leads to a robust antitumor immune response.

**Figure 3 cells-14-01627-f003:**
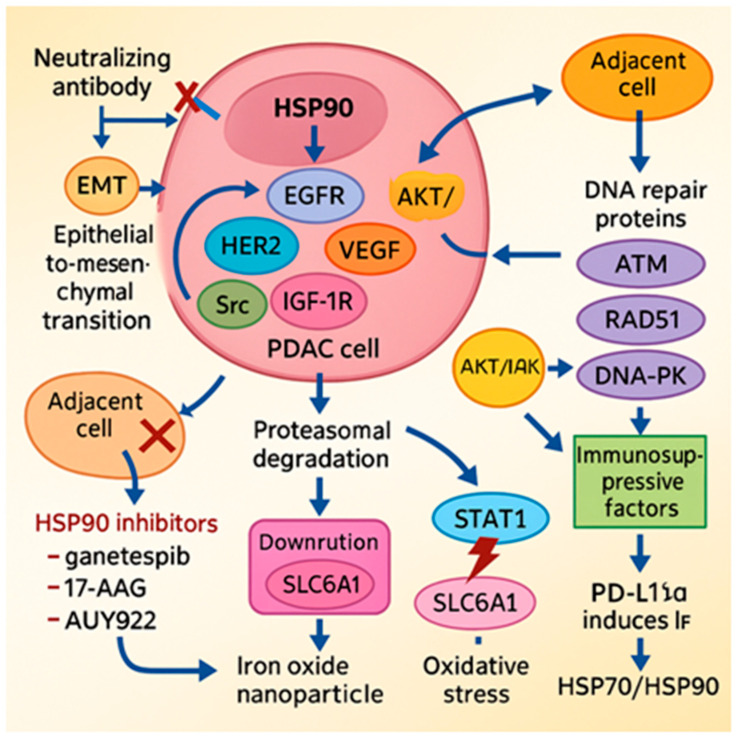
The role of HSP90 in PDAC progression and potential therapeutic strategies. This diagram illustrates the key functions of Heat Shock Protein 90 (HSP90) within a cancer cell and potential pharmacological interventions. Inside the PDAC cell, HSP90 stabilizes oncogenic “client” proteins such as EGFR (epidermal growth factor receptor), HER2 (human epidermal growth factor receptor 2), Src (proto-oncogene tyrosine-protein kinase Src), IGF-1R (Insulin-like Growth Factor 1 Receptor), VEGF (Vascular Endothelial Growth Factor), and the AKT signaling pathway, which all promote cancer cell survival and proliferation. HSP90 also promotes EMT (epithelial-to-mesenchymal transition). The figure also shows interactions with adjacent cells and the DNA repair pathway, involving proteins like ATM (ataxia-telangiectasia mutated), RAD51 (recombinase RAD51), and DNA-PK (DNA-dependent protein kinase). The figure also highlights therapeutic strategies targeting HSP90: HSP90 inhibitors, including ganetespib, 17-AAG, and AUY922, which block protein activity. Neutralizing antibodies that inhibit the EMT process. Iron oxide nanoparticles which affect SLC6A1 (Solute Carrier Family 6 Member 1) expression. Synergistic effects of inhibitors with other pathways, such as the STAT1 (Signal Transducer and Activator of Transcription 1) pathway and PD-L1α (programmed death-ligand 1 alpha), leading to suppression of immunological response. All these processes and strategies are the subject of research in laboratory settings (in vitro) using cell lines and in studies on animal models of cancer (in vivo).

**Table 1 cells-14-01627-t001:** The role of HSPs in pancreatic cancer pathogenesis. This table summarizes the pathological roles of different heat shock protein families in the development and progression of the disease. The “HSP Family” column identifies specific protein families, including ImHSPs, HSP40 (DnaJ Family), HSP60 (Chaperonins), HSP70, HSP90, and HSP100. The “Pathological Role in PDAC” column details how these proteins contribute to the tumor’s pathogenesis, such as by inhibiting apoptosis, promoting invasiveness, increasing chemoresistance, and inducing epithelial–mesenchymal transition (EMT) and cachexia. The “References” column provides the numbers of scientific publications that are the source of the presented information. Abbreviations used in the table are defined as follows: PDAC (pancreatic ductal adenocarcinoma), BiP/GRP78 (binding immunoglobulin protein/glucose-regulated protein 78), OXPHOS (oxidative phosphorylation), ERK1/2 (extracellular signal-regulated kinases 1 and 2), p38MAPK (p38 mitogen-activated protein kinase), CAFs (cancer-associated fibroblasts), Nrf2 (nuclear factor erythroid 2-related factor 2), GPx (glutathione peroxidase), and MMP2/9 (matrix metalloproteinase 2 and 9).

HSP Family	Pathological Role in PDAC	References
lmHSPs	Ferroptosis inhibition, promoting chemoresistance via Snail/E-cadherin/ERCC1 (HSP27); tumor suppression via p53 (HSPB2)	[25,28,35,58,59]
HSP40 (DnaJ Family)	Promoting PDAC development via BiP/GRP78 (DnaJB11); apoptosis inhibition, promoting invasiveness, enhancing Warburg effect and Bcl-2 expression (DnaJA1)	[23,33,34]
HSP60 (Chaperonins)	Apoptosis inhibition via HSP60/OXPHOS/Erk1/2 pathway; overexpression correlates with PDAC severity	[15,35,37]
HSP70	Promoting EMT via NF-κB; development of cachexia via p38βMAPK; overexpression in tumor cells and CAFs (HSPA2)	[15,21,41,42,43,44,45,46,60,61]
HSP90	Ferroptosis resistance via Nrf2/GPx; mutant p53 stabilization; inducing invasiveness via MMP2/9 activation; promoting EMT and immune evasion	[21,48,49,52,62,63]
Large HSPs (HSP100)	Unknown	[21,39,48,64]

**Table 2 cells-14-01627-t002:** The importance of HSPs as diagnostic and prognostic markers in pancreatic cancer. This table provides an overview of the diagnostic and prognostic implications of various heat shock proteins in PDAC. The column titled “Marker (HSP)” lists specific HSP family members, such as HSPB6, HSPB1 (HSP27), and HSP90. The “Diagnostic/Prognostic Relevance” column describes how the expression of each HSP is correlated with disease progression. The “Methods/Models” column specifies the experimental approaches used to obtain these findings, which include mass spectrometry analysis of human tumor cell lines (in vitro) and studies using mouse models (in vivo). The “References” column provides the citations for the data.

Marker (HSP)	Diagnostic/Prognostic Relevance	Methods/Models	References
HSPB6	Overexpressed in cancer-associated fibroblasts (CAFs); associated with improved overall survival in patients with PDAC; prognostic marker in PDAC	Mass spectrometry analysis of cancer-associated fibroblasts and cancer cell lines (Clinical Proteomic Tumor Analysis Consortium)	[27]
HSPB1 (HSP27)	Lower expression linked to poor overall survival in patients with PDAC after resection and liver metastases; higher expression associated with a better response to gemcitabine in the resected, non-metastasized patient group	Immunoreactive score (IRS), post-resection PDAC patient data (Dexter et al.)	[72]
HSP90	High levels indicate poor prognosis; in PET imaging the expression of this protein enables monitoring and early detection of pancreatic cancer	PET radiotracers, mouse model, immunochemistry (Wang et al.); pathologic data (Gamboa et al.); mice and rat models (Kacar et al.)	[53,54,56,57]

**Table 3 cells-14-01627-t003:** Selected experimental and pharmacological strategies targeting HSPs in pancreatic Ccncermodels. This table summarizes different compounds and experimental strategies used to inhibit HSPs in cancer research. The “Strategy/Compound” column lists agents such as triptolide, melatonin, maslinic acid, and a combination of RFA (radiofrequency ablation) + mTOR (mammalian Target of Rapamycin) inhibitors. The “Mechanism of Action” column explains how each strategy works, such as inducing apoptosis, suppressing metastasis, or enhancing chemotherapy sensitivity. The “Targeted HSPs” column identifies the specific HSP family members that are affected by each agent. These are experimental strategies that have been studied in both in vitro (on cell lines) and in vivo (on animal models) settings.

Strategy/Compound	Mechanism of Action	Targeted HSPs	References
Triptolide (TPL)	HSF1 inhibition and caspase-3, caspase-9 degradation promotes apoptosis and leads to increased tumor sensitivity to chemotherapy	HSP27, HSP70, HSP90	[65]
Active hexose-correlated compound (AHCC)	Gemcytabine/methylglyoxal pathway leads to overexpression of HSP27, which is downregulated by AHCC inducing apoptosis and preventing resistance to chemotherapy	HSP27	[71]
siRNA + ATRA delivered by PEGylated polyethylenimine-coated gold nanoparticles	HSP47-specific mRNA degradation by siRNA prevents ECM proliferation and increases gemcytabine sensitivity	HSP47	[75]
AK-778, Col003, Pirfenidon	Direct inhibition of HSP47 inhibits tumor growth and increases gemcytabine sensitivity	HSP47	[74]
Local hyperthermia	Increases tumor antigenicity and drug penetration by enhancing HSP70 and HSP60 expression; HSP70 promotes antitumor immune response, while HSP60 activates T cells and IFN-γ secretion	HSP60, HSP70	[76]
Metformin + aminoguanidine	GLO-1 inhibition interferes with methylglyoxal/HSP27/HSP70 pathway increasing PDAC sensitivity to gemcytabine	HSP27, HSP70	[68]
Melatonin	HSP27, HSP60, HSP70, HSP90, and HSP100 downregulation via NF-κB and STAT3 inhibition promotes apoptosis and increases tumor sensitivity to chemotherapy	HSP27, HSP60, HSP70, HSP90, HSP100	[71]
Melatonin, AFMK	Suppression of HSP70 and cIAP-2 enhances gemcitabine efficacy and promotes apoptosis	HSP70	[82]
Ap-4-139B + Hydroxychloroquine	Selective HSP70 inhibition induces mitochondrial swelling and activates the apoptotic pathway; combination with hydroxychloroquine (autophagy inhibitor) enhances antitumor efficacy	HSP70	[77]
Pancastatin A and B	GRP78 (HSPA5) inhibition during glucose deprivation	HSP70	[85]
Xanthone derivative of secalonic acid D	AKT signaling pathway inhibition under glucose-starved condition and GRP78 (HSPA5) downregulation leads to cytotoxic activity on PANC-1	HSP70	[86]
Maslinic acid	Proliferation inhibition and inducing autophagy in PANC-28 through HSPA8 downregulation	HSP70	[78]
DIO-NPs + HSP Inhibitors	DIO-NPs induce cellular stress leading to increased HSP70/HSP90 expression; combination with HSP inhibitors may impair survival mechanisms of PDAC and enhance therapy efficacy	HSP70, HSP90	[99]
RFA + mTOR Inhibitors	Inhibition of RFA-induced via HSP70 AKT/mTOR pathway leads to suppression of proliferation and enhanced therapeutic response	HSP70	[100]
JG-231	Mortalin (HSPA9, GRP75) inhibition in K-RasG12C mutation PDAC increases the permeability of the mitochondrial membrane and promotes apoptosis	HSP70	[65]

## Data Availability

No new data were created or analyzed in this study.

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
