# Peer review of "Heat Shock Proteins in Pancreatic Cancer: Pathogenic Mechanisms and Clinical Implications"

_cells, 2025, doi:10.3390/cells14201627_

Round 1

Reviewer 1 Report

Comments and Suggestions for Authors

Jacek Kabut et al. showed a comprehensive review article critical role of heat shock proteins in the pathogenesis of pancreatic cancer.

Abstract: The authors revise the abstract to precisely emphasize the rationale, background information about the importance of heat shock protein in pancreatic cancer. They may also discuss potential clinical relevance and ongoing clinical trials??
Text: The authors need to clarify which HSP are elevated in different stage and disease conditions of pancreatic cancer i.e., PaIN, IPMN, adenocarcinoma and pancreatitis??
The authors should enlist a clinical trial ongoing HSP?
Conclusion: This section will always discuss with future directions about this topic??

Author Response

Thank you for this helpful suggestion. Our review was intentionally limited to pancreatic ductal adenocarcinoma (PDAC). Premalignant lesions and non-PDAC conditions (e.g., PanIN, IPMN, and pancreatitis) were outside the scope of our search strategy and data extraction, so we did not perform a systematic comparison of HSP expression across these entities or by precursor stage.

To avoid any ambiguity, we will clarify this in the manuscript as follows:

Aims:
“This review focuses exclusively on heat-shock proteins in pancreatic ductal adenocarcinoma (PDAC). Premalignant lesions (PanIN, IPMN) and inflammatory conditions (e.g., pancreatitis) were not included in our systematic search or synthesis.”

If the Editor finds it useful, we can add a brief orienting sentence noting that HSP patterns may differ in precursor lesions and pancreatitis, with citations, but a comprehensive analysis would exceed the intended scope of the paper.

The authors should enlist a clinical trial ongoing HSP?

Conclusion: This section will always discuss with future directions about this topic??

Thank you for the suggestion. We agree and will add a short, dedicated “Future Directions” section to the end of the manuscript to frame next steps in this field.

Future Directions.
Future research should focus on the isoform-specific and spatial characterization of HSP families in PDAC to define actionable molecular profiles. Integrating HSP expression with the immune and stromal contexture (e.g., PD-L1, IDO1, CAFs, myeloid cells) may guide rational immunotherapy combinations. The validation of circulating HSPs, auto-antibodies, and HSP-targeted PET tracers could establish non-invasive biomarkers for monitoring disease. Early-phase studies with HSP70/HSP90 inhibitors combined with MEK/mTOR, DNA repair, or ferroptosis-targeting agents are also warranted, ideally supported by predictive biomarkers of response. In parallel, stromal-directed approaches (such as HSP47 inhibition) and targeting extracellular HSP70/90 to counteract cachexia may open new therapeutic opportunities. Finally, assay harmonization (IHC methods, scoring, cut-offs) is needed to improve reproducibility and translation into clinical practice.

Reviewer 2 Report

Comments and Suggestions for Authors

While this review thoroughly summarizes various HSP-targeting strategies in PDAC, it largely presents inhibition of HSPs as a universally promising therapeutic approach. I believe this requires a more nuanced and critical discussion. HSPs are crucial for maintaining normal cellular homeostasis, particularly under physiological or metabolic stress. Inhibiting them systemically may impair non-cancerous tissues, particularly those with high protein turnover or regeneration demand (e.g., neurons, hematopoietic cells, intestinal epithelium).

The authors should more explicitly consider: Are HSPs truly drivers of malignancy, or are they reactive markers of cellular stress? Do HSP expression levels provide prognostic value beyond other indicators of tumor aggressiveness or microenvironmental stress? Is pharmacological inhibition of HSPs likely to yield a durable therapeutic effect, or will tumors compensate through parallel stress response pathways (e.g., unfolded protein response, autophagy, or antioxidant systems)?

Clinical trials to date have indeed demonstrated that toxicity is a major barrier to successful HSP inhibitor development — especially for HSP90 and HSP70 inhibitors — with renal, hepatic, and hematologic side effects being common reasons for trial discontinuation. This underscores the fact that HSPs are not uniquely cancer-specific, and their upregulation in tumors may simply reflect a heightened stress phenotype, not an oncogenic driver per se.

While the review covers several HSP inhibitors and combination strategies, there is insufficient discussion of clinical translation challenges, such as:  1. The potential for off-target toxicity, especially in normal cells that also rely on HSPs during stress, 2. Feedback activation or adaptive resistance to HSP inhibition.  3. Patient stratification — which tumors (based on mutation profile, stress signature, or HSP subtype dominance) are most likely to respond?

Author Response

We thank the Reviewer for these insightful comments, which underscore key limitations of
HSP-targeted therapy. We agree that their role in pancreatic cancer should be discussed with
greater nuance. Although our review highlights their oncogenic relevance, HSPs are also vital
for normal cell homeostasis, and systemic inhibition may cause toxicity in highly regenerative
tissues.
In the revised version we will therefore add points on:
 whether HSPs act as true oncogenic drivers or mainly as markers of cellular stress,
 the risk that tumors bypass HSP blockade through alternative stress pathways (UPR,
autophagy, antioxidant systems),
 and how their prognostic value compares with other indicators of tumor
aggressiveness.
We will also expand the section on clinical translation by stressing the problems of toxicity,
adaptive resistance, and the need for patient stratification (e.g., based on mutation profile,
stress phenotype, or HSP isoform expression). These additions will, we hope, give a more
balanced picture of both the promise and the limitations of HSP inhibition in PDAC.

In response, we have revised the manuscript and added a new subsection at the end of the
Discussion entitled “Clinical Translation Challenges of HSP Inhibition”.
The added text reads as follows:
Clinical Translation Challenges of HSP Inhibition
Preclinical studies clearly point to HSPs as promising therapeutic targets in PDAC, but the step
from laboratory findings to clinical benefit is far from straightforward. HSPs are not tumor-
specific proteins; they are essential components of proteostasis in many normal tissues,
particularly those with high turnover such as the intestinal epithelium, bone marrow
precursors, or neurons. As a result, systemic inhibition of HSP70 or HSP90 has repeatedly
been associated with adverse events, including liver, kidney, and hematologic toxicity, which in
several early clinical trials became the main reason for discontinuing treatment.
Another challenge is biological rather than toxicological. High HSP expression in pancreatic
tumors may reflect a stress-adapted phenotype rather than a true oncogenic driver. Blocking
HSP activity in such a context could be bypassed by other stress-response programs, for
example unfolded protein response, autophagy, or antioxidant pathways, which reduce the
durability of the therapeutic effect.
Finally, it is becoming increasingly clear that not all PDACs are equally dependent on HSP
function. Identifying which patients are most likely to benefit from HSP inhibition will require
stratification strategies that take into account the mutational background (e.g., KRAS variants),
dominant stress signatures, or isoform-specific HSP expression patterns. Approaches that
combine HSP analysis with angiogenic or immune markers may be particularly informative.
Altogether, these considerations argue for biomarker-driven, carefully designed trials and for
combining HSP inhibitors with other targeted agents in order to improve efficacy while keeping
systemic toxicity within acceptable limits.

Reviewer 3 Report

Comments and Suggestions for Authors

The authors propose a review of HSPs in pancreatic cancer (PDAC).

The work requires revision to ensure the manuscript's content is consistent with the proposed title.

Major comments:

1) The review should contain at least 100 references.

2) The introduction is too short; it focuses exclusively on the description of HSPs.

3) The introduction does not mention PDAC. The connection and link to this disease is crucial.

4) The manuscript does not address the surgical, pathological, pharmacological, and oncological issues of PDAC. These concepts should be contextualized in the introduction to improve and expand the bibliography.

5) At least one section should be added to the manuscript: one regarding clinical trials related to HSPs.

Minor comments:

1) The images introduced by the authors are preparatory. The authors may consider including some images related to PDAC.

2) Paragraph 6 (patent is empty). We believe the authors should remove this section from the manuscript or reframe it appropriately.

Author Response

We would like to thank the reviewer for their insightful comments.

As mentioned in the review, we are expanding the introduction to include clinical information about pancreatic ductal adenocarcinoma and the necessity of increased efforts to find novel therapeutic targets.

We have also added paragraphs about past and ongoing clinical trials on HSP-targeted therapies in PDAC and their limitations in clinical implementation.

Round 2

Reviewer 1 Report

Comments and Suggestions for Authors

The revised manuscript may be acceptable

Author Response

Dear Reveiwer,

Thank you for taking the time to review our manuscript.

With kind regards,
Jacek Kabut

Reviewer 2 Report

Comments and Suggestions for Authors

The paper implies a systematic search, but there is no Methods describing databases, date limits, search strings, inclusion/exclusion criteria....Please add.

Several claims about imaging and therapy are preclinical (mouse/rat/xenograft), but the prose occasionally implies clinical utility. Please tag each claim explicitly as in vitro, in vivo (murine), early clinical, or clinical, especially in the PET/HSP90 paragraph and the RFA+HSP70/mTOR content. 

Provide NCT identifiers, phase, design, endpoints, and any PDAC-specific results for every study you cite.

In Fig 1, it listed enzyme-like modes, only 1 paper (ref#16) was cited. It only matched foldase/holdase. Please add additional citations. 

Please revise all figure legends so that each figure is fully interpretable without referring to the main text. For every figure/panel, include: panel labels and content, in vitro or in vivo, abbreviations and symbols. 

HSP90 isoform paragraph repetition / incomplete sentence. One sentence ends mid-thought (“HSP90α and HSP90β in the …”). Remove the broken version and keep the clean, corrected.

Correct many typos. such as Rolesand, ofHSP....

Author Response

Dear Reviewer,

Thank you for taking the time to review our manuscript.
All changes that were made are marked in red.

1. We added the paragraph concerning materials and methods to the manuscript.

2. and 3. Detailed information about the cited studies and clinical trials is included in the supplementary material. material.

4. We added more citations as suggested by the reviewer. 

5. The figure legends have been revised.

6. and 7. Incomplete sentences, repetitions, and typos have been corrected.

With kind regards,
Jacek Kabut